# UNBOUNDED: A GENERATIVE INFINITE GAME OF CHARACTER LIFE SIMULATION

**Jialu Li**[1,2*]    **Yuanzhen Li**[1]    **Neal Wadhwa**[1]    **Yael Pritch**[1]    **David E. Jacobs**[1]
**Michael Rubinstein**[1]    **Mohit Bansal**[2]    **Nataniel Ruiz**[1]
[1]Google    [2]The University of North Carolina at Chapel Hill
**https://infinite-generative-game.github.io/**

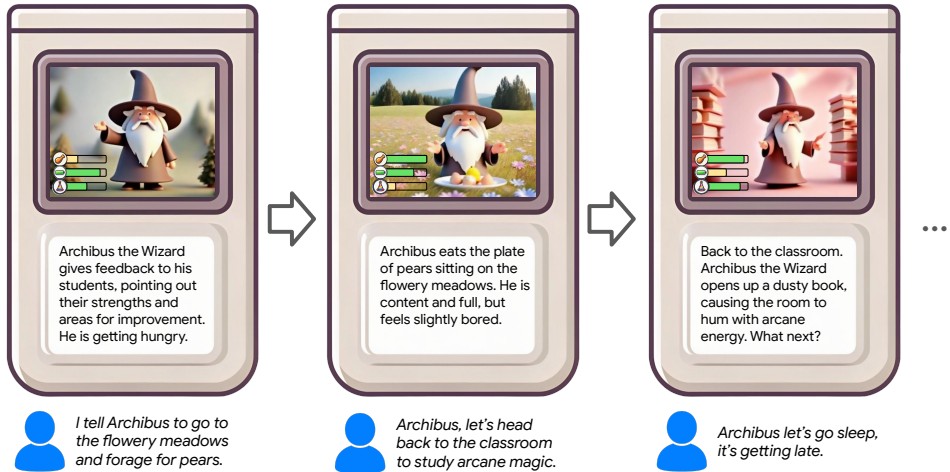

Figure 1: An example of UNBOUNDED. We follow the life of *Archibus*, the user's custom wizard character. The user can interact with the generative game using natural language, and Archibus' hunger, energy and fun meters update accordingly. A spontaneous and unconstrained story unfolds while the user playing, and the character can explore new environments with a myriad of possible actions and unexpected interactions. The game runs in interactive speeds, refreshing every second.

## ABSTRACT

We introduce the concept of a **generative infinite game**, a video game that transcends the traditional boundaries of finite, hard-coded systems by using generative models. Inspired by James P. Carse's distinction between finite and infinite games (Carse, 1986), we leverage recent advances in generative AI to create UNBOUNDED: a game of character life simulation that is fully encapsulated in generative models. Specifically, UNBOUNDED draws inspiration from sandbox life simulations and allows you to interact with your autonomous virtual character in a virtual world by feeding, playing with and guiding it - with open-ended mechanics generated by an LLM, some of which can be emergent. In order to develop UNBOUNDED, we propose technical innovations in both the LLM and visual generation domains. Specifically, we present: (1) a specialized, distilled large language model (LLM) that dynamically generates game mechanics, narratives, and character interactions in real-time, and (2) a new dynamic regional image prompt Adapter (IP-Adapter) for vision models that ensures consistent yet flexible visual generation of a character across multiple environments. We evaluate our system through both qualitative and quantitative analysis, showing significant improvements in character life simulation, user instruction following, narrative coherence, and visual consistency for both characters and the environments compared to traditional related approaches.

---

[*]Work done during an internship at Google

# 1    INTRODUCTION

In his work *"Finite and Infinite Games: A Vision of Life as Play and Possibility"* Carse (1986), James P. Carse introduces a distinction between two types of games. Carse defines **finite games** as those *"played for the purpose of winning,"* with boundaries, fixed rules, and a definitive endpoint. In contrast, **infinite games** are *"played for the purpose of continuing the play,"* with no fixed boundaries and evolving rules. Traditional video games are, inherently, finite games due to the limitations of computer programming and computer graphics. For example, all game mechanics have to be fully pre-defined in the programming language and all graphics assets have to be pre-designed (modulo procedural generation which still grapples with structural limitations). This allows for only a finite, and sometimes predefined, set of actions and paths that can be taken. They also feature predefined rules, boundaries, and win conditions.

Recent advances in generative models have been impressive. We hypothesize that these developments have finally opened up the possibility of creating the first generative infinite video game. Two advancements have made this achievable: (1) large language models (LLMs) that can encode persistent video game mechanics (e.g. interactions with the game environment or characters, character state tracking, object permanence), generate interactive stories, and produce spontaneous, and sometimes emergent, behaviors; and (2) visual generative models capable of producing high-quality images that follow prompts. In this work, we present UNBOUNDED, what we believe to be the first interactive generative infinite game, where game behaviors and outputs are generated by AI models, transcending the constraints of hard-coded systems. UNBOUNDED draws inspiration from sandbox life simulations and digital pet games such as Little Computer People, The Sims and Tamagotchi. It incorporates elements from tabletop roleplaying games like Dungeons & Dragons, which offer unconstrained storytelling experiences that have been unattainable in video games.

Previous work in game generation includes fluidic games merging gameplay with design Gaudl et al. (2018), automated game design (Cook et al., 2016), and AI-focused games (Treanor et al., 2015b). These emphasize exploration within predefined game parameters, unlike our vision of generative infinite games, which evolve beyond fixed structures with real-time content and mechanics generation. Other efforts, like Agarwal (2024), Sun et al. (2023), and (Liapis et al., 2019), generate parts of games, but not all components. In contrast, in our work, all of the game mechanics, characters, environments, narrative, and graphics are fully produced by generative models. This is more similar in spirit to recent work Genie (Bruce et al., 2024) and GameNGen (Valevski et al., 2024), although in contrast to these works that mainly generate platformers with diffuse mechanics or regenerate behaviors of one pre-existing game, our work proposes an open-ended narrative experience with stable game mechanics enabled by an LLM-based game engine.

UNBOUNDED offers a gameplay loop centered around character simulation and open-ended interaction (as shown in Figure 11 in Appendix). Players can insert their characters into the game, defining their appearance and personality. The game generates a world where these characters can explore environments, interact with objects, and engage in conversations. The game generates new scenarios, stories, and challenges based on the player's actions and choices, creating a personalized and infinite gaming experience.

Specifically, UNBOUNDED has the following capabilities: (1) **Character Personalization**: players can insert their characters into the game, defining their appearance and personality. (2) **Game Environment Generation**: UNBOUNDED generates a persistent world that the characters can explore and interact. (3) **Open-Ended Interaction**: Players can interact with the character using natural language instructions, and there are no pre-defined rules to constraint the interaction. (4) **Real-Time Generation**: we pay special attention to the speed of the game and achieve 5-10x speedups over a naive implementation, serving each new scene with a latency of about *one second*.

Our approach introduces technical innovations in both LLM and vision generation domains. On the language side, we developed a **LLM based game engine** capable of maintaining consistent game mechanics, generating coherent narratives, and producing contextual character responses in real-time. Our distilled specialized model is fine-tuned on data automatically generated with two collaborative strong LLM agents, without the need for human annotation in the loop. Our distilled LLM model handles the dynamic generation of game rules and scenarios, adapting to player input and game state. In visual generation, we introduce a new **regional IP-Adapter**, which allows for the consistent generation of characters and environments while maintaining visual coherence across

multiple images. Specifically, our regional IP-Adapter conditions the image generation on the game environment and character appearance encoded modulated by a dynamic mask obtained from attention outputs in cross-attention layers. This is in order to mitigate the interference between the environment and character, in order to have both reliably appear in the scene. This approach enables real-time image generation that reflects the game state and player actions.

The contributions of this work are conceptual and technical. We introduce the notion of a generative infinite game, demonstrating its feasibility and potential impact on the future of interactive entertainment. We present a new paradigm for game design where the game logic and content are encapsulated within generative models. Our main technical contributions include the specialized distilled LLM for game logic and narrative generation and the regional IP-Adapter for consistent visual generation. We demonstrate the effectiveness of our regional IP-Adapter through both quantitative and qualitative evaluations, surpassing state-of-the-art in both character and environment consistency. Furthermore, we show that our distilled LLM performs comparably to a very large LLM while having interactive speed. These advancements enable the creation of UNBOUNDED and lay the groundwork for future research and development in the field of AI-driven interactive experiences.

## 2 RELATED WORK

**Controllable Text-to-Image Generation.** Controllable text-to-image generation becomes a key research direction in diffusion model applications, enabling diverse ways to guide the generation process. For instance, ControlNet (Zhang et al., 2023) introduces conditioning mechanisms that utilize control signals such as depth maps, poses, edges, and segmentation maps to guide image generation. Other works focus on layout control using bounding boxes to control object placement within the generated images (Li et al., 2023; Shin et al., 2022). Beyond these control signals, another major area of research involves personalization, where the goal is to generate consistent characters (Ruiz et al., 2023; Gal et al., 2022; Kumari et al., 2023) or consistent face identities (Li et al., 2024b; Wang et al., 2024d; Yan et al., 2023; Ruiz et al., 2024; Jeong et al., 2023) across multiple generations. However, most existing approaches lack support for conditioning both characters and environments separately (Tewel et al., 2024; Gal et al., 2024; Avrahami et al., 2024; Wei et al., 2023; Liu et al., 2024; Tao et al., 2025; He et al., 2024; Zhou et al., 2024d; Wang et al., 2024e;c;b; Shentu et al., 2024; Wang et al., 2024d). Others may require predefined masks or layouts for generating characters, with the environment remaining identical to the input image (Chen et al., 2024; Lugmayr et al., 2022; Yang et al., 2023). This limitation makes it difficult to seamlessly integrate characters into different environments while ensuring both consistency and alignment with the input prompts. IP-Adapter (Ye et al., 2023) tackles this task by conditioning the generation on the environment and character images. However, IP-Adapter tends to over reconstruct the conditions, which causes interference between them. In this paper, we build our approach on IP-Adapter and propose an improved regional IP-Adapter with block drop, separating character and environment generation to enhance consistency.

**Game Generation.** In the field of Procedural Content Generation (PCG), diverse approaches have been explored to create dynamic game content (Summerville et al., 2018; Shaker et al., 2016) or game rules Khalifa et al. (2017), including concept maps (Treanor et al., 2012), conceptual expansion (Guzdial & Riedl, 2021; 2018), Markov Chains (Snodgrass & Ontañón, 2014), Bayes Nets (Guzdial & Riedl, 2016), and LSTMs (Summerville & Mateas, 2016). Recent research has advanced to using GANs for generating game levels or dynamic environments (Volz et al., 2018; Kumaran et al., 2019; Schubert et al., 2021; Kim et al., 2020), diffusion models for game generation (Zhou et al., 2024b; Sun et al., 2023), LLMs to design and generate the game environments or mechanics (Sudhakaran et al., 2024; Todd et al., 2023; Nasir & Togelius, 2023; Hu et al., 2024; Zala et al., 2024; Anjum et al., 2024; Chung & Kreminski, 2024; Chung et al., 2024). While AI typically aids in generating individual game components in previous works, our work aims to fully generate all game behaviors, including graphics, characters, environments, and narrative, using generative models.

Bruce et al. (2024) synthesize interactive games using a video diffusion model based on previous frames and user actions, but their approach doesn't involve infinite games. We argue that generating compelling infinite games by modeling pixels alone is challenging, so we use language models for

open-ended game mechanics. Valevski et al. (2024) train a diffusion model to be a game engine running Doom, but it focuses on a single, finite game that already exists.

Besides generating the game content, there has been exploration in letting AI take different roles (e.g., competitor, designer, or teammate) in games (Treanor et al., 2015a; Zhu et al., 2021; Gallotta et al., 2024; Pell, 1992; Agarwal et al., 2023), and system for automated game design on grids (Cook, 2022). Another line focuses on generating narrative text games or using AI tools to generate or act in games (Ammanabrolu et al., 2020; Li et al., 2024a; Latitude Inc., 2023; Wang et al., 2023a; Zhu et al., 2023; Kreminski et al., 2020; Cui et al., 2023; Zhou et al., 2023; Dambekodi et al., 2020). Our work differs by (1) generating all game mechanics, graphics, characters, environments, and narrative, (2) adding personalization with custom characters and story arcs, and (3) achieving real-time interactivity through innovations in vision and language.

**Large Language Models in Image Generation.** Large language models have demonstrated strong in-context learning capability (Brown, 2020), which enables them to solve diverse customized tasks based on human instructions and in-context examples. In the field of image generation, large language models have been employed for various tasks, such as image layout generator based on prompts (Cheng et al., 2024a;b), interactive multi-turn image generation (Zeqiang et al., 2023; Huang et al., 2024; Gong et al., 2023; Wang et al., 2023b), and interleaved text and image generator (Team, 2024; Zhou et al., 2024a). Unlike these applications, our work focuses on distilling a specialized LLM, serving as a game engine, responsible for generating game mechanics, narratives and character interactions.

## 3 METHOD

We introduce UNBOUNDED, an interactive generative infinite game powered by text-to-image generation models and large language models. UNBOUNDED offers: (1) **Custom Character Personalization**: users create unique characters with customizable appearances and personalities; (2) **Dynamic World Creation**: the system generates a persistent, interactive game world for exploration; (3) **Open-Ended Interaction and Gameplay**: players interact with their characters via natural language, with the game dynamically generating new scenarios and storylines based on player actions; and (4) **Generation in Interactive Speed**: the game runs with near real-time interactivity, achieving a refresh rate close to *one second*. We detail the methods enabling these capabilities in this section.

### 3.1 PERSONALIZATION OF LATENT CONSISTENCY MODELS FOR CHARACTER CONSISTENCY

A key feature of UNBOUNDED is its ability to serve a fully generative model-based game with real-time interaction. This is made possible through the use of latent consistency models (LCM) (Luo et al., 2023) which allow for high-resolution image generation with as few as two diffusion steps. By utilizing LCMs, UNBOUNDED achieves real-time text-to-image (T2I) generation, critical for delivering an interactive gaming experience with a refresh rate close to *one second*.

To support the use of custom characters in the game, we incorporate DreamBooth (Ruiz et al., 2023) into the T2I model. Given a set of character images, we fine-tune the diffusion model using LoRA modules (Hu et al., 2021). During fine-tuning, we append a unique identifier, "[V]", which has a weak prior in the model, to denote the subject. DreamBooth personalization is performed on the base diffusion model, and we merge the subject-specific LoRA with the LCM LoRA trained for few-step diffusion. This simple arithmetic LoRA merging works surprisingly well in maintaining both inference speed and subject preservation. Other alternatives for personalization exist, and many do not need setup time, yet we find that they often fail in strongly preserving the character's features, which is a critical component in order to have a satisfactory experience in this type of game.

### 3.2 REGIONAL IP-ADAPTER WITH BLOCK DROP FOR ENVIRONMENT CONSISTENCY

Another key feature of UNBOUNDED is generating the character in pre-defined environments performing different actions based on user instructions. Thus, maintaining both character and environment consistency is essential. While character consistency is handled as discussed in Sec. 3.1, two additional challenges arise: ensuring the environment consistency across different generations and

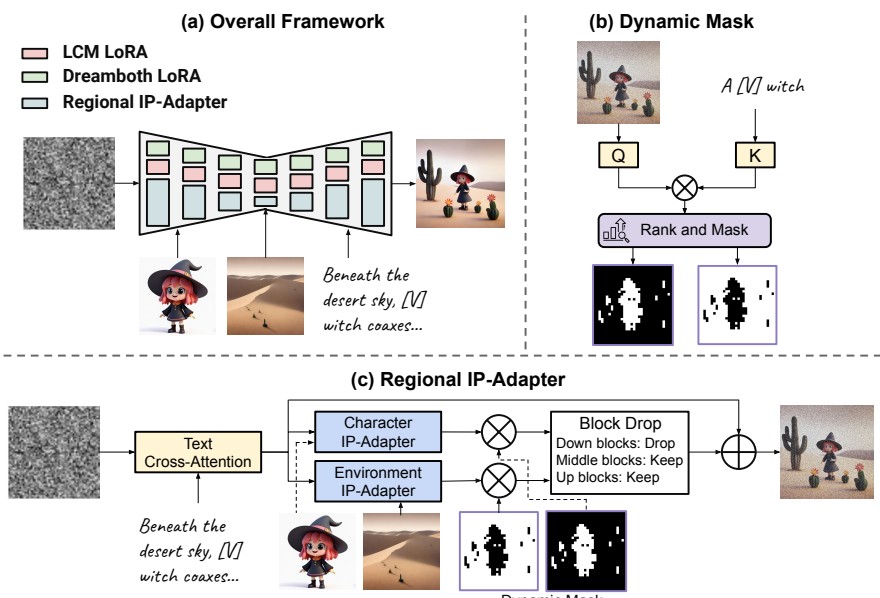

Figure 2: (a) Our overall image generation method. We achieve real-time image generation with LCM LoRA, maintain character consistency with DreamBooth LoRAs, and introduce a regional IP-Adapter (shown in (c)) for improved environment and character consistency. (b) Our proposed dynamic mask genreation separating the environment and character conditioning, preventing interference between the two.

accurately placing the character within the environment, without losing alignment with text prompts. We find that existing method fails to consistently perform well for all requirements in interactive speed. As one of our main technical contributions we propose a novel **regional IP-Adapter** in order to consistently implant a character in pre-defined environments following text prompt.

### 3.2.1 REGIONAL IP-ADAPTER

We propose an improved version of IP-Adapters (Ye et al., 2023) that enables dual-conditioning on both subjects and environments, allowing for the generation of a pre-defined character in a user-specified environment. Unlike the original IP-Adapters, which focus on single-image conditioning, our approach introduces dual-conditioning and dynamic regional injection mechanism to represent both concepts simultaneously in the generated images.

Let us start with an example. As shown in Figure 2, given the text prompt *"Beneath the desert sky, [V] witch coaxes the cacti to blossom with vibrant, glowing flowers"* and the desert environment image, the model needs to know that the character should be generated beside cacti and within the desert environment. This requires the model to correctly (1) preserve the environment (2) preserve the character (3) follow the prompt. Utilizing IP-Adapter to encode the environment greatly harms both (2) and (3) (Figure 4). Our regional IP-Adapter solves this by implementing a novel attention separation mechanism for generating the two elements. Specifically, we introduce a dynamic mask-based approach that leverages cross-attentions between the character text embedding and the hidden states at each layer of the model. As shown in Figure 2, our approach applies the adapter to the regions corresponding to the environment and character separately, preventing the environment conditioning to interfere with the character's appearance, and vice versa. At each cross-attention layer, we calculate the attention map between a pre-defined character text embedding $K_c$ and the output hidden states from the text cross-attention layer $O_t$:

$$A_c = \frac{W_q O_t * W_k K_c^T}{\sqrt{d}} \qquad (1)$$

where $W_q$ and $W_k$ are projection weights adopted from the text cross-attention layers. The dynamic mask we use for the regional IP-Adapter is defined as:

$$M_c = \begin{cases} 1 & A_c \leq threshold \\ 0 & A_c > threshold \end{cases} \quad (2)$$

We rank the attention scores $A_c$ and set the $threshold$ at top $r\%$, dynamically updating the mask at each attention layer throughout the diffusion process. The output of the cross-attention blocks is calculated as:

$$O = O_t + \alpha_e M_c * O_e + \alpha_c(1 - M_c) * O_c \quad (3)$$

where $O_e$ and $O_c$ represent the outputs of the IP-Adapter image cross-attention layers of environment and character, and $\alpha_e$ and $\alpha_c$ are the IP-Adapter scales to adjust the strength of the environment and character conditioning respectively. Our regional IP-Adapter with dynamic masks allows the model to generate the character without the interference from the environment conditioning, greatly enhancing character consistency preservation (Figure 4). Besides, it preserves environment consistency by generating the background around the character conditioning on the environment image, while maintaining the implicit layout information in the hidden states from the text cross-attention layers $O_t$, ensuring the character is placed accurately following the text prompt.

### 3.2.2 BLOCK-WISE DROP ON ENVIRONMENT CONDITIONING

For our regional IP-Adapter, we use a dynamic mask from the cross-attention between character text and hidden states. This mask's quality is key for separating character and environment generation. Figure 12 in Appendix shows attention maps between character embeddings and hidden states in cross-attention layers of down sample blocks. We observe that the attention doesn't focus on the character but spreads across the full image for these blocks. We take this as a strong indication that the diffusion model doesn't separate character and environment generation in these layers and instead focuses on overall image structure based on text prompts. This aligns with Wang et al. (2024a)'s finding that down sample blocks capture spatial layouts more, while up sample blocks capture style. We drop the regional IP-Adapter in down sample blocks, using it only in mid and up sample blocks. This allows for better spatial layout generation between character and environment. Additionally, we find that adding dynamic regional IP-Adapters in up sample blocks more strongly aligns the generated background with the conditioning environment by extracting relevant semantic information while preserving character-specific details like appearance and poses.

### 3.3 LANGUAGE MODEL GAME ENGINE WITH OPEN ENDED INTERACTIONS AND INTEGRATED GAME MECHANICS

UNBOUNDED simulates character actions in pre-defined environments with images generated based on scene text descriptions while monitoring the character's state and providing the user with natural language interaction with the character and game environment. For example, the user can take the character to different environments, interact with the character (e.g. *"I pet the character on its head"*), or essentially take any open ended action e.g. *"I pet the character and then take them for a rocket ride at the space station."*. Since the game is ultimately built on a language model, these expansive capabilities present several challenges: (1) *Environment Binding*: the model needs to place the character in the correct environment based on the natural language instructions. (2) *Coherent Story Generation*: the model generates coherent narrative descriptions and character responses that align with user-specified character traits. (3) *Game Mechanics Generation*: the model needs to monitor the state of the character (i.e., hunger, energy, fun, hygiene), and update them based on user interactions and story events. (4) *Prompt Rewriting*: the model needs to rewrite the narratives for the diffusion model (i.e., append special token "[V]" for the character, align environment descriptions to the pre-generated environments for better environment consistency).

Surprisingly, we find that a very large language model (e.g., GPT-4, GPT-4o) with detailed instructions and using in-context learning (Brown, 2020) can exhibit these capabilities. Nevertheless, using such large models as game engines is not directly feasible due to the large latency (e.g., 5 seconds for a 7B model to give one response). Given this, we propose to distill these capabilities from a very large model into a smaller model based on Gemma-2B (Team et al., 2024) for game logic and narrative generation that supports real-time interaction. In this section we propose two key technical

contributions (1) a design for a character life simulation game using two very large language models that control for world modeling and user interaction respectively (2) a framework for distilling this knowledge into one smaller language model that is fast enough to achieve interactive speeds.

### 3.3.1 CHARACTER LIFE SIMULATION WITH MULTI-LLM COLLABORATION

We build a character life simulation game with two LLMs agents. One agent serves as the world simulation model, responsible for setting up game environments, generating narratives and image descriptions, tracking character states and simulating the character's behavior. The second agent functions as a user model, simulating the player's interactions with the world simulation model. It has three types of interactions: continuing the story within the current environment, moving the character into different environments, or interacting with the character to maintain the healthy state of the character. In each interaction category, the user has the option to provide personality details of the character or guide the character actions that, in turn, guide the simulator's narrative generation. This interaction between the world simulation LLM and the user LLM allows for a dynamic character life simulation game with virtually unlimited interaction possibilities and narrative paths.

### 3.3.2 FRAMEWORK FOR SMALL LLM DISTILLATION

We propose a framework for distilling the capabilities of larger LLMs into the smaller, more efficient model using synthetic data generated by the multiple stronger LLMs. Our framework contains two steps: (1) automated data collection and (2) small model distillation.

**Automated Data Collection** Our goal is to build a general-purpose character life simulator capable of understanding a wide range of character traits and generating games across diverse topics. To achieve this, the first step is to gather a diverse dataset of topics and characters. We prompt a large LLM to generate pairs of topics and corresponding main characters. To ensure data diversity, we retain only the generated pairs whose ROUGE-L similarity to existing data is below 0.7, following (Wang et al., 2022), which demonstrated the importance of diverse data for enhancing an LLM's ability to follow instructions. This process results in 5,000 unique topic-character pairs, which serve as the basis for user-simulator interaction data.

In the second step, we collect multi-round interaction data between the world simulation LLM and the user LLM. The process begins with the world simulation LLM setting up the game environment and initiating a character action based on a randomly sampled topic-character pair from the dataset. The user LLM is then prompted to provide interaction inputs, while the world simulation LLM generates updated character actions, states, and responses. This iterative process continues for five interaction rounds per session, resulting in a total of 5,000 user-simulator interaction examples. All the prompt templates are in Appendix.

**Distillation** Once the interaction data has been collected, we fine-tune the smaller Gemma-2B model using the 5,000 synthetic user-simulator interaction samples. During supervised fine-tuning, we mask out the loss on user input data, focusing the optimization on learning the world simulation model's behavior based on multi-round interaction history and current user input. This approach enables Gemma-2B to replicate the capabilities of larger LLMs as a game engine while supporting real-time interaction. Our distilled Gemma-2B demonstrates performance comparable to GPT-4o, effectively following user input and supporting unbounded interactions.

## 4 EXPERIMENTAL SETUP

### 4.1 EVALUATION BENCHMARKS

**Evaluation of Image Generations** To evaluate our image generation approach, we collect an evaluation dataset consisting of 5,000 (character image, environment description, text prompt) triplets with GPT4o (OpenAI, 2023). It includes 5 characters (dog, cat, panda, witch, and wizard), 100 diverse environments, and 1,000 text prompts (10 per environment). We evaluate the image generation performance under three criteria: environment consistency, character consistency and semantic alignment with the text prompt. To measure similarity between images, we employ CLIP-I (Radford et al., 2021), DINO (Caron et al., 2021), and DreamSim (Fu et al., 2023). We denote the

Table 1: Comparison of UNBOUNDED and other methods for maintaining environment consistency and character consistency. UNBOUNDED achieves the best performance in maintaining consistency, while maintaining comparable semantic alignment with the text prompt. Best scores are in **bold**.

| Methods | Environment Consistency | | | Character Consistency | | | Semantic Alignment |
|---|---|---|---|---|---|---|---|
| | CLIP-I$^E$ ↑ | DINO$^E$ ↑ | DreamSim$^E$ ↓ | CLIP-I$^C$ ↑ | DINO$^C$ ↑ | DreamSim$^C$ ↓ | CLIP-T ↑ |
| IP-Adapter (Ye et al., 2023) | 0.470 | 0.381 | 0.595 | 0.366 | 0.139 | 0.832 | 0.168 |
| IP-Adapter-Instruct (Rowles et al., 2024) | 0.334 | 0.151 | 0.832 | 0.246 | 0.124 | 0.872 | 0.098 |
| StoryDiffusion (Zhou et al., 2024c) | 0.528 | 0.257 | 0.733 | 0.629 | 0.464 | 0.545 | **0.242** |
| Ours | **0.563** | **0.322** | **0.675** | **0.676** | **0.470** | **0.488** | **0.242** |

Table 2: Ablations of the effectiveness of dynamic regional IP-Adapter and block drop in our consistent image generation approach.

| No. | Block Drop | Regional IP-Adapter | Scale | Environment Consistency | | | Character Consistency | | | Alignment |
|---|---|---|---|---|---|---|---|---|---|---|
| | | | | CLIP-I$^E$ ↑ | DINO$^E$ ↑ | DreamSim$^E$ ↓ | CLIP-I$^C$ ↑ | DINO$^C$ ↑ | DreamSim$^C$ ↓ | CLIP-T ↑ |
| 1. | ✗ | ✗ | 1.0 | 0.123 | 0.111 | 0.885 | 0.073 | 0.024 | 0.973 | 0.034 |
| 2. | ✓ | ✗ | 1.0 | 0.414 | **0.331** | **0.647** | 0.337 | 0.147 | 0.832 | 0.149 |
| 3. | ✓ | ✓ | 1.0 | **0.563** | 0.322 | 0.675 | **0.676** | **0.470** | **0.488** | **0.242** |
| 4. | ✗ | ✗ | 0.5 | 0.470 | **0.381** | **0.595** | 0.366 | 0.139 | 0.832 | 0.168 |
| 5. | ✓ | ✗ | 0.5 | **0.577** | 0.332 | 0.640 | 0.627 | 0.374 | 0.575 | **0.252** |
| 6. | ✓ | ✓ | 0.5 | 0.549 | 0.263 | 0.726 | **0.705** | **0.514** | **0.450** | 0.246 |

similarity between environment reference image and the generated images as CLIP-I$^E$, DINO$^E$, DreamSim$^E$, and the similarity between the character reference image and the generated images as CLIP-I$^C$, DINO$^C$, DreamSim$^C$. Additionally, we use CLIP-T (Radford et al., 2021) to evaluate the semantic alignment with the text prompt. Given that UNBOUNDED is a character life simulation game, ensuring the presence of the character in the image is important. Therefore, we further utilize Grounding-DINO (Liu et al., 2023) to detect the presence of the character in the generated images. We set similarity scores to 0 and distance scores to 1 if there is no character in the generated image.

**Evaluation of LLM Generations**   We collect an additional evaluation dataset with 100 user-simulator interaction samples using the pipeline in Sec. 3.3. Each user-simulator interaction sample contains five rounds of interaction between the user and the world model. We use GPT-4 (OpenAI, 2023) as a judge, scoring the response between two models (baseline model vs. our model) in overall score, and then in four aspects: accuracy of character state update, environment relevance, story coherence, and user input instruction following. Scores range from 0 to 10.

## 4.2 IMPLEMENTATION DETAILS

Our image generator is built on SDXL (Podell et al., 2023). We train a DreamBooth LoRA of rank 16 with batch size 1 and a constant learning rate 1e-4 for 500 steps on a single A100, which takes approximately 30 mins. The special token we append before the character is "sks". During inference, we merge the LCM-LoRA with DreamBooth LoRA with scale 1.0 for each. We use IP-Adapter-plus-sdxl-vit-h for encoding the environment, and IP-Adapter-plus-face-sdxl-vit-h for encoding the character. The dynamic mask ratio $r\%$ in set to be 60%.

Our LLM is built on Gemma-2B (Team et al., 2024). We distill the LLM using 5,000 user-simulator interaction samples collected from GPT-4o. We train the LLM for 6,500 steps, with batch size 8, distributed across 4 A100s, and learning rate 1e-4. The learning rate scheduler is set to be cosine annealing (Loshchilov & Hutter, 2016), and the warmup steps ratio is 0.03. In evaluation, we use an LLM to generate responses with sampling. The sampling hyperparameters are set to be default.

## 5 RESULTS AND ANALYSIS

### 5.1 COMPARISON WITH DIFFERENT APPROACHES FOR MAINTAINING ENVIRONMENT CONSISTENCY AND CHARACTER CONSISTENCY

**Quantitative Results** We compare our regional IP-Adapter with block drop with previous approaches in maintaining environment consistency and character consistency. For all the approaches, we merge the character LoRA and LCM LoRA with the model to support fast inference and im-

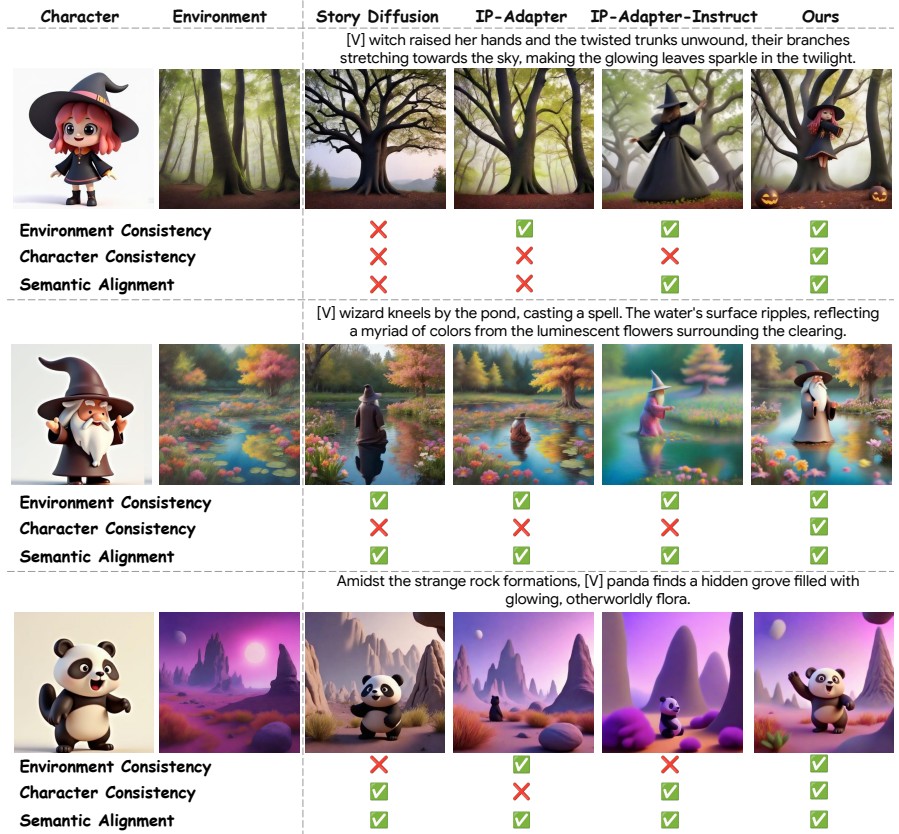

Figure 3: Comparison with other approaches for generating environment and character consistent images based on text prompts. We observe that our method strongly outperfoms related work.

prove character consistency and provide an apples-to-apples comparison. As shown in Table 1, our approach consistently outperforms previous approach in maintaining environment consistency and character consistency, while achieving comparable performance in maintaining semantic alignment. Specifically, our approach significantly overtakes StoryDiffusion (Zhou et al., 2024c) by 0.047 in CLIP-I$^C$, and 0.057 in DreamSim$^C$ for character consistency, and 0.035 in CLIP-I$^E$, 0.065 in DINO$^E$, and 0.058 in DreamSim$^E$ for environment consistency, demonstrating the effectiveness of our approach. Besides, our approach also achieves comparable performance in maintaining semantic alignment, suggesting strong text following capabilities.

**Qualitative Results** We present a qualitative comparison with other approaches in Figure 3. Our regional IP-Adapter with block drop consistently generates images with high character consistency, whereas other methods may fail to include the character or generate characters with inconsistent appearances (see Example 1 & 2). Furthermore, we show that our approach balances environment consistency and character consistency well, while other approaches might generate environments that differ from the condition environment (e.g., StoryDiffusion in Example 1 & 3).

## 5.2 EFFECTIVENESS OF DYNAMIC REGIONAL IP-ADAPTER WITH BLOCK DROP

**Quantitative Results** We demonstrate that our regional IP-Adapter with block drop is essential for placing the character in the environments following the text prompt, while maintaining both environment and character consistency with ablation studies. As shown in Table 2, adding block drop improves both environment and character consistency compared with multi-IP-Adapter (No. 2. vs. No. 1.), with an increase of 0.291 in CLIP-I$^E$ and 0.264 in CLIP-I$^C$, alongside better alignment between the text prompt and the generated image. Furthermore, our regional IP-Adapter enhances character consistency and text alignment while maintaining comparable performance in environment consistency (No. 3 vs. No. 2). We also explore the effect of the environment IP-Adapter scale, and

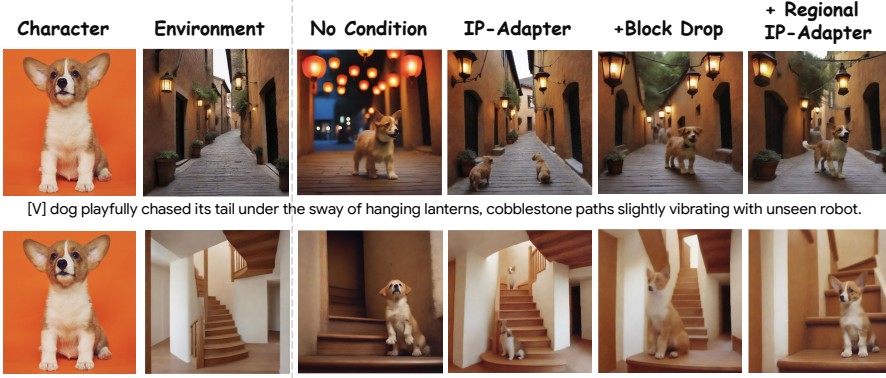

Figure 4: Comparison between our regional IP-Adapter approach and baseline approaches.

Table 3: Comparison of UNBOUNDED and different LLMs on serving as game engines for open ended interactions and integrated game mechanics. We use GPT-4 to provide pairwise scores between our model and other LLMs.

| Model | Overall | | State Update | | Environment Relevance | | Story Coherence | | Instruction Following | |
|---|---|---|---|---|---|---|---|---|---|---|
| | Base | Ours | Base | Ours | Base | Ours | Base | Ours | Base | Ours |
| Gemma-2B (Team et al., 2024) | 6.22 | **7.44** | 5.60 | **7.47** | 6.12 | **7.94** | 6.34 | **7.57** | 6.43 | **7.67** |
| Gemma-7B (Team et al., 2024) | 6.80 | **7.39** | 6.29 | **7.43** | 7.07 | **7.91** | 6.90 | **7.48** | 6.89 | **7.53** |
| Llama3.2-3B (Meta, 2024) | 7.21 | **7.50** | 6.86 | **7.38** | 7.63 | **7.93** | 7.36 | **7.56** | 7.31 | **7.67** |
| Ours-1k | 7.65 | **7.82** | 7.50 | **7.74** | 8.10 | **8.19** | 7.78 | **7.93** | 7.82 | **7.97** |
| GPT-4o (OpenAI, 2023) | **7.76** | 7.68 | **7.69** | 7.66 | **8.20** | 8.10 | **7.95** | 7.82 | **7.85** | 7.82 |

our findings indicate that using a smaller scale (e.g., 0.5) generally improves character consistency, though it slightly compromises environment consistency (No. 6 vs. No. 3).

**Qualitative Results** As shown in Figure 4, conditioning on the environment using IP-Adapter achieves good environment reconstruction, but the character consistency is influenced by the environment style. Introducing block drop improves adherence to the text prompt, resulting in images with the correct spatial layout for both the character and the environment. However, the character's appearance remains influenced by the surrounding environment. By incorporating our proposed regional injection mechanism with our proposed dynamic mask scheme, the generated images achieve strong character consistency while maintaining effective conditioning on the environment.

## 5.3 EFFECTIVENESS OF DISTILLING SPECIALIZED LARGE LANGUAGE MODEL

We show that our diverse user-simulator interaction data effectively distills Gemma-2B into a capable game engine. As shown in Table 3, zero-shot inference with small LLMs (i.e., Gemma-2B, Llama3.2-3B), or a slightly larger LLM (i.e., Gemma-7B) results in lower performance compared to ours, highlighting the importance of distillation from a stronger LLM for game world and character action simulation. Furthermore, we show that our model achieves performance comparable to GPT-4o, validating the effectiveness of our approach. We also investigate the impact of distillation data size on performance by comparing a Gemma-2B model distilled with 1K data and 5K data. Results show that using a larger dataset consistently improves performance across all aspects, highlighting the potential for further enhancements with more data to fully match GPT-4o's performance.

## 6 CONCLUSION

We introduce UNBOUNDED, an interactive generative infinite game based on generative models. UNBOUNDED is built on two main components, a specialized, distilled LLM for real-time interaction, and a fast diffusion model with our proposed regional IP-Adapter for consistent generation across multiple scenes. We show that our proposed approach allows for an interactive game subsumed in generative models, with consistent characters, environments and story and an expansive gameplay characteristic of an infinite game.

## ACKNOWLEDGMENTS

We thank Shiran Zada, Peyman Milanfar, Shlomi Fruchter, Michael Goin and Matthew Guzdial for the thoughtful feedback and discussion.

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

APPENDIX

In this appendix, we present the following:

- A qualitative example of our UNBOUNDED in Sec. A.
- Human evaluation results for LLM as game engine in Sec. B.
- Human evaluation results for dynamic regional IP-Adapter in Sec. C.
- Evaluation of LLM as game engine with multiple samples in Sec. D.
- A pipeline figure for our UNBOUNDED in Figure 11, and the attention figure in UNet when extracting dynamic mask in Figure 12.
- Prompts we use for user-simulation data collection in Sec. E.
- Evaluation prompts we use for querying GPT-4 as a judge for LLM evaluation in Sec. F.
- Reproducibility statement in Sec. G.

## A    QUALITATIVE EXAMPLE OF UNBOUNDED

In this section, we include one qualitative example of UNBOUNDED. The example was collected using GPT-4o as the user interacting with our game engine, with no interference or cherry-picking of favorable outcomes. As shown in Figures 7, 8, 9, 10, given an input game topic, UNBOUNDED generates relevant and diverse environments aligned with the theme. Besides, when the user interacts with the character, the character responds naturally (e.g., the witch's eyes light up with excitement as she says, "Let's see what this puzzle holds!"). Furthermore, the character's state updates dynamically throughout the game. As the game progresses, the character's states—such as hunger, energy, fun, and hygiene—decline, reflecting the character growing hungry, tired, less entertained, or dirty. Users can freely engage with the character to restore these states by feeding them, playing, letting them rest, or encouraging them to bathe (e.g., the user says, "I show the witch a celestial puzzle to solve for extra fun," and the character's fun level increases). Lastly, we observe emergent behaviors during gameplay. For instance, when the user stated, "The witch is adventurous and courageous, eager to uncover secrets and delve into the unknown realms that ancient wisdom holds. Please continue the generation accordingly," the game engine autonomously took the witch to explore a new environment, interpreting it as the destination connected to the ancient tome's hidden passage. This example provides an overview of how the game operates and demonstrates how unbounded interactions can lead to unique and dynamic game outcomes.

## B    HUMAN EVALUATION FOR LLM AS GAME ENGINE

In this section, we present a human evaluation to assess the performance of our distilled LLM as a game engine. Specifically, we sample 10 user-simulator interaction examples using the pipeline described in Sec. 3.3. Each interaction consists of five rounds of exchanges between the user and the world model. Responses from the world model are generated using both GPT-4o and our distilled small LLM. We compare the

Table 4: Human Evaluation on 50 responses from LLM on 10 game topics. We show the detailed win/lose/tie counts on each evaluation category.

| Criteria | Win (%) | Lose (%) | Tie (%) |
|---|---|---|---|
| Overall Score | 26.5 | 27.0 | 46.5 |
| State Update | 20.5 | 24.0 | 55.5 |
| Environment Relevance | 16.0 | 15.5 | 68.5 |
| Story Coherence | 15.0 | 18.0 | 67.0 |
| Instruction Following | 14.5 | 14.5 | 71.0 |

responses from the two models across a total of 50 paired examples based on the dialogue history and ask four human annotators to evaluate them across five categories: overall score, state update accuracy, environment relevance, story coherence, and adherence to user input instructions. The user interface employed for collecting human annotations is shown in Figure 5. To eliminate bias, the response order between the two models is randomized, ensuring that annotators are unaware of which model generated a given response. Table 4 details the annotators' votes for win, lose, and tie outcomes for UNBOUNDED and GPT-4o. Our distilled small LLM demonstrates performance comparable to GPT-4o across all evaluated categories.

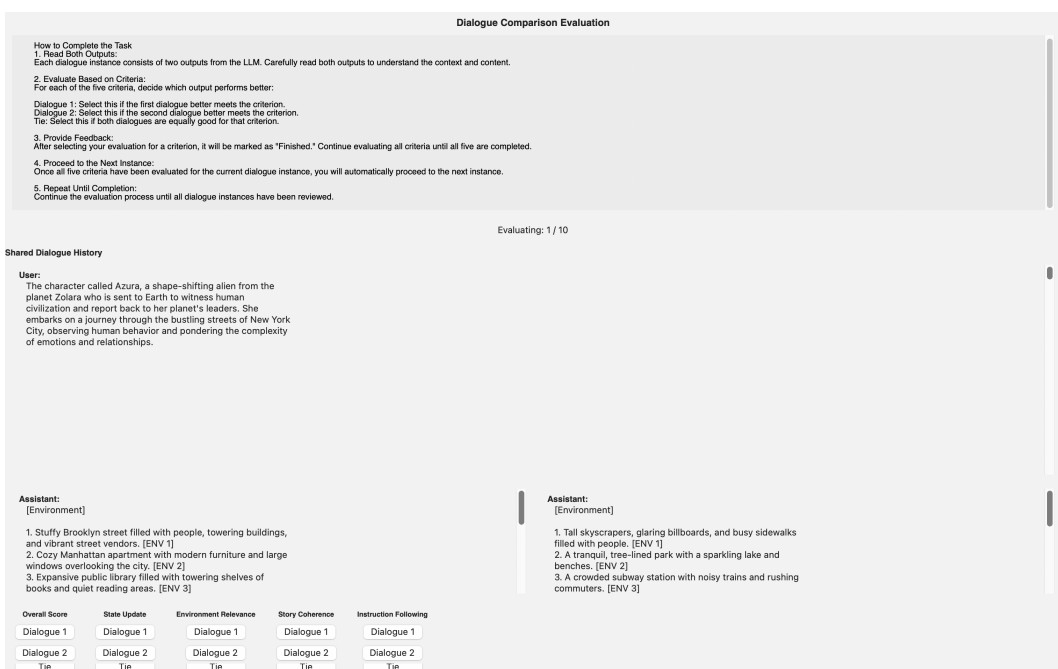

Figure 5: Example user interface for human evaluation for LLM as game engine.

## C HUMAN EVALUATION FOR IMAGE GENERATION

In this section, we present a human evaluation to assess the performance of our dynamic regional IP-Adapter in maintaining environment consistency, character consistency, and semantic alignment with the text prompt. Specifically, we sample 50 character-environment-text prompt triplets from our image evaluation benchmark and generate corresponding images using our approach and StoryDiffusion Zhou et al.

Table 5: Human Evaluation on 50 generated images from our dynamic regional IP-Adapter and StoryDiffusion. We show the detailed win/lose/tie counts on each evaluation category.

| Criteria | Win (%) | Lose (%) | Tie (%) |
|---|---|---|---|
| Character Consistency | 60.0 | 19.0 | 21.0 |
| Environment Consistency | 54.0 | 21.0 | 25.0 |
| Semantic Alignment | 57.5 | 23.0 | 19.5 |

(2024c). Four human annotators were tasked with evaluating the images across three dimensions: character consistency, environment consistency, and semantic alignment. Given that UNBOUNDED functions as a character life simulation game, ensuring the presence of the character in the image is critical. To emphasize this, annotators were instructed to disfavor images where the character is absent. In such cases, the image is also rated lower in terms of environment consistency and semantic alignment. The user interface used for collecting human annotations is shown in Figure 6. To minimize bias, the order of images generated by the two models was randomized, ensuring that annotators were unaware of the model responsible for generating each image. Table 5 summarizes the annotators' votes for win, lose, and tie outcomes for the dynamic regional IP-Adapter and StoryDiffusion. Our evaluation demonstrates that the dynamic regional IP-Adapter significantly outperforms StoryDiffusion in maintaining character consistency, environment consistency, and semantic alignment with the text prompt.

## D EVALUATION OF LLM AS GAME ENGINE WITH MULTIPLE SAMPLES

We evaluated our LLM using two decoding strategies—greedy decoding and sampling—to obtain more robust comparisons. Greedy decoding generates a consistent output for identical inputs, reducing randomness in evaluation. The results are presented in Table 6. We observed that greedy decoding outperforms GPT-4o when evaluated using GPT-4 as the judge. Additionally, we con-

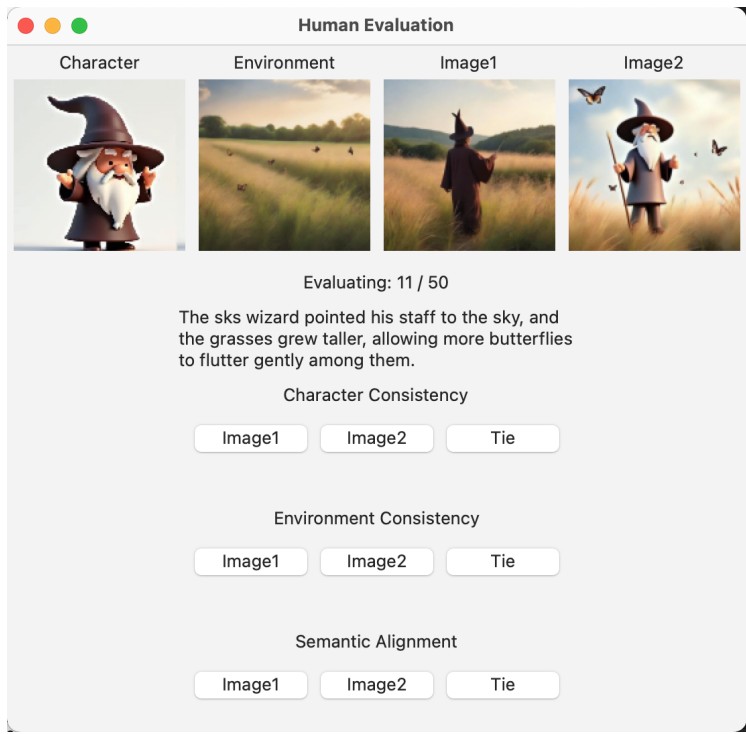

Figure 6: Example user interface for human evaluation for our dynamic regional IP-Adapter.

ducted three more runs using sampling (samples 2, 3, and 4) and included the results here. Despite the variations in responses across different samples, all outputs maintain high quality under the evaluation criteria.

Table 6: Evaluation results comparing our model with GPT-4o under greedy and sampled decoding. Scores are judged by GPT-4 on five metrics: Overall, State Update, Environment Relevance, Story Coherence, and Instruction Following.

| Sample | Overall | | State Update | | Environment Relevance | | Story Coherence | | Instruction Following | |
|---|---|---|---|---|---|---|---|---|---|---|
| | Ours | GPT-4o | Ours | GPT-4o | Ours | GPT-4o | Ours | GPT-4o | Ours | GPT-4o |
| Greedy | **7.93** | 7.44 | **7.83** | 7.20 | **8.27** | 7.62 | **8.04** | 7.62 | **8.05** | 7.60 |
| Sample1 | 7.68 | **7.76** | 7.66 | **7.69** | 8.10 | **8.20** | 7.82 | **7.95** | 7.82 | **7.85** |
| Sample2 | **7.50** | 6.94 | **7.40** | 6.69 | **7.76** | 7.08 | **7.60** | 7.08 | **7.64** | 7.04 |
| Sample3 | **7.87** | 7.38 | **7.89** | 7.12 | **8.24** | 7.68 | **7.92** | 7.52 | **8.01** | 7.51 |
| Sample4 | **7.61** | 7.16 | **7.56** | 6.90 | **7.90** | 7.47 | **7.69** | 7.28 | **7.72** | 7.30 |

# E    PROMPT FOR SYNTHETIC USER-SIMULATOR INTERACTION DATA COLLECTION

In this section, we provide the prompt templates we use to collect the user-simulation data. Specifically, we query GPT-3.5 to generate diverse topics and character descriptions using the template shown in Figure 13. The prompt template for guiding the potential user interactions for user LLM is shown in Figure 14. To ensure user interactions align with the dialogue history, we include the interaction history as input to the user LLM. The world simulation LLM prompt template, shown in Figure 15, also takes in the dialogue history and generates the next character actions, states and narratives. We constrain the world simulation LLM to generate one storyline at a time, allowing users to choose how to continue the story.

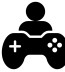 A game of the witch: Step into a starry nightscape at a celestial observatory perched high above the clouds. Use astrological magic to manipulate the stars and planets, crafting spells that can influence time itself. Solve cosmic puzzles and unlock ancient knowledge hidden among the constellations.

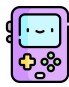 **Environments**

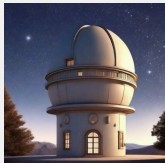 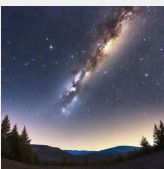 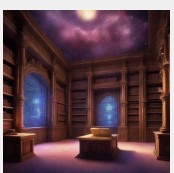 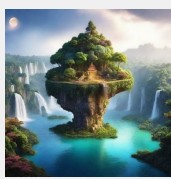 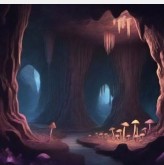

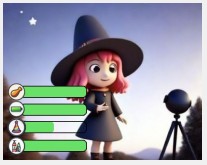 witch stands at the observatory's telescope, gazing at the interconnected stars.

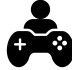 I show the witch a celestial puzzle to solve for extra fun.

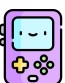 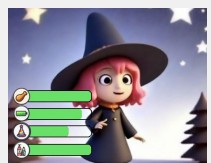 witch examines the celestial puzzle, its stars glimmering with potential solutions.
[Character Response]
The witch's eyes light up with excitement and says: "Let's see what this puzzle holds!"

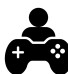 I bring the witch to the ancient library within a nebula.

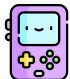 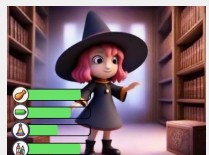 witch steps into the ancient library, marveling at the mystical tomes and celestial artifacts.

Figure 7: Qualitative Example of Unbounded (Split 1). The example is collected with GPT-4o as the user to interact with our game.

## F    EVALUATION PROMPT

We include the prompt template we use to compare the outputs from two LLMs in Figure 16. The prompt is adapted from Vicuna (Chiang et al., 2023), and has been validated as an effective tool for comparing the performance of two LLMs on a given task.

## G    REPRODUCIBILITY STATEMENT

First, we include the implementation details of UNBOUNDED in Sec. 4.2, covering the training hyperparameters for Dreambooth fine-tuning, and LLM distillation, and the hyperparameters we use during inference for both LLM and image generation. Second, we provide detailed description of the user-simulation data we collect for training in Sec. 3.3, and further include the prompt template used to query GPT models in Appendix Sec. E. Lastly, for the LLM-based evaluation, the prompt template for querying GPT-4 to compare two LLM outputs is provided in Appendix Sec. F.

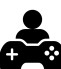 The witch is curious and intellectual, always drawn to the mysteries and knowledge hidden in ancient texts. Please continue the generation accordingly.

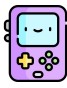 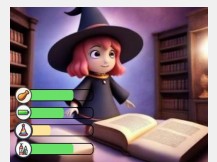 witch carefully pulls out a dusty, ancient tome, its cover glowing with celestial energy.

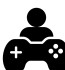 The witch has a strong affinity for celestial magic, often feeling invigorated by the cosmic forces within mystical artifacts. Please continue the generation accordingly.

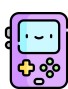 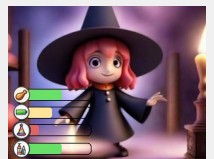 witch concentrates, casting spells from the mystical tome, feeling the cosmic forces invigorate.

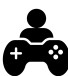 I offer the witch a magical snack infused with cosmic energy to satisfy hunger.

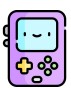 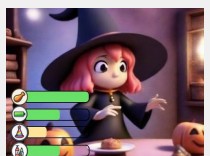 witch enjoys the magical snack, feeling rejuvenated and ready to delve deeper into the tome.
[Character Response]
The witch gratefully accepts the snack, saying: "Thank you, this will give me the strength to continue."

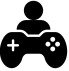 I guide the witch through a playful, magical exercise to boost her fun.

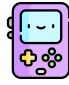 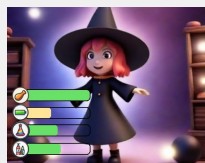 witch dances with the celestial artifacts, feeling a surge of playful, magical energy.
[Character Response]
The witch laughs and says: "This is delightful! Let's see what rewards these new skills!"

Figure 8: Qualitative Example of Unbounded (Split 2).

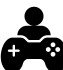 The witch has a lively and imaginative spirit, thriving on the energy of playful magic and celestial exploration. Please continue the generation accordingly.

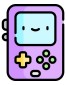 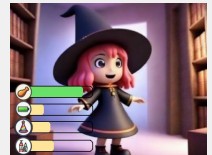 witch discovers a hidden passage in the ancient tome.

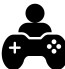 The witch is adventurous and courageous, eager to uncover secrets and delve into the unknown realms that ancient wisdom holds. Please continue the generation accordingly.

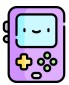 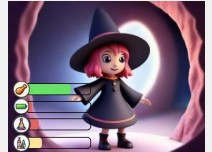 witch steps through the hidden passage, finding herself in a swirling portal to a celestial realm.

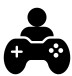 I encourage the witch to rest and regain energy before venturing further.

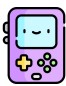 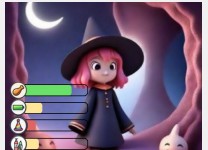 witch finds a serene spot in the celestial realm to rest, feeling the cosmic energies revive.
[Character Response]
The witch nods and says: "A rest will refresh my mind. Thank you for the suggestion."

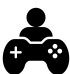 I bring the witch to the floating island with magical gardens.

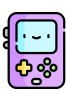 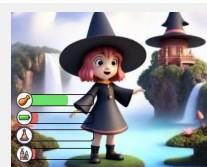 witch arrives on the floating island, marveling at the magical gardens and sparkling waterfalls of light.

Figure 9: Qualitative Example of Unbounded (Split 3).

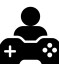 I offer the witch a refreshing bath beneath the waterfalls of light to rejuvenate and improve hygiene.

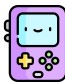 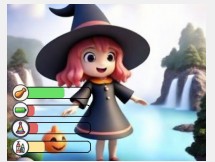 witch splashes in the refreshing waters, feeling rejuvenated and her hygiene improved.
[Character Response]
The witch smiles gratefully and says: "A bath sounds wonderful. Thank you!"

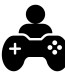 I bring the witch to the vast, star-studded sky with cosmic puzzles.

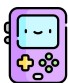 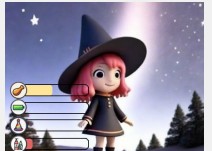 witch steps into the vast, star-studded sky, feeling the ancient cosmic puzzles calling to her.

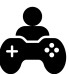 I provide the witch with a nourishing meal to alleviate her hunger and restore energy.

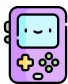 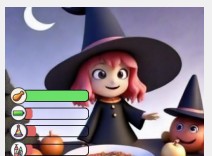 witch enjoys the nourishing meal, feeling her energy and spirit revive.
[Character Response]
The witch eagerly eats the meal and says: "This is exactly what I needed. Thank you!"

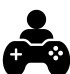 I engage the witch with a delightful cosmic game to boost her fun.

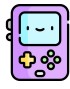 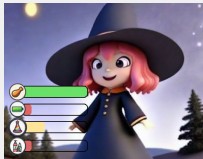 witch joyfully participates in the delightful cosmic game, feeling her fun levels skyrocket.
[Character Response]
The witch laughs joyfully and says: "This is so much fun! I feel completely engaged."

Figure 10: Qualitative Example of Unbounded (Split 4).

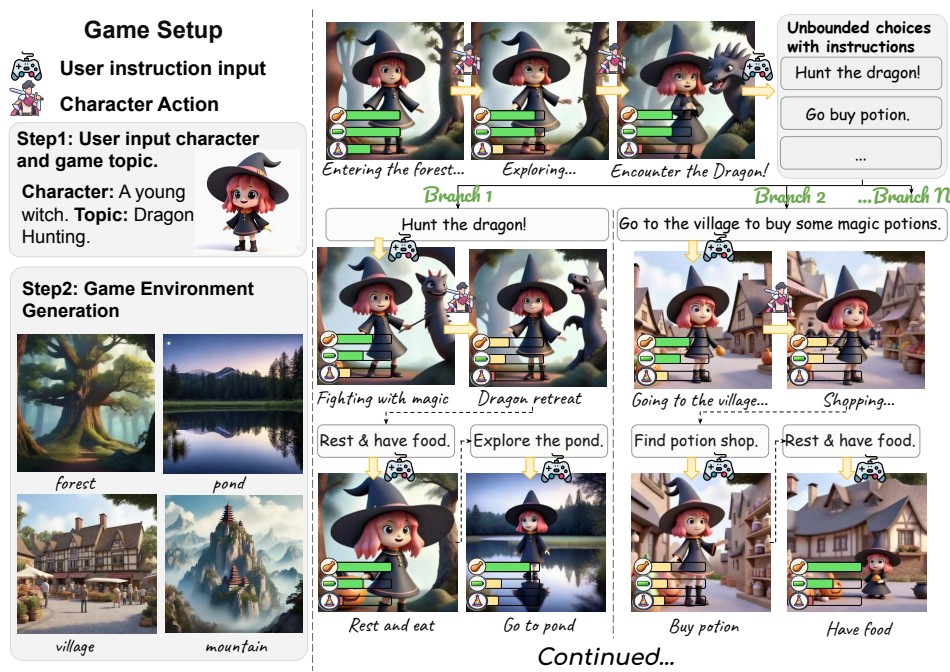

Figure 11: Example of UNBOUNDED. Based on an initial user input, UNBOUNDED sets up game simulation environments, and generates character actions in the environments. Users can interact with the character with natural language instructions, exploring the game with unlimited options.

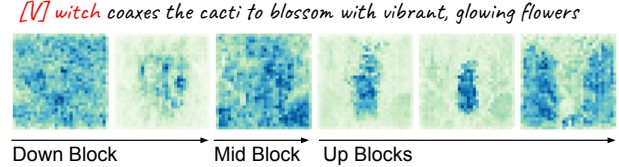

Figure 12: Attention map between character embedding and hidden states in cross-attention layers in different blocks. The character embedding we use is "A [V] witch".

**Topic and Character Collection**

You're an agent that is responsible for generating the character and a story topics for the character. Here're several instructions you need to follow:

1. You should come up with a character and a story topic, and you should pair them together.
2. The character should be diverse enough to cover human, animal, robot.
3. The topic should be as very high-level (e.g., space adventure, daily life, adventure to the candy garden).
4. The character and story topic should be as diverse as possible.
5. You should generate the character and story topic in one natural language sentence. Don't split the character description and story topic in two lines.
6. Please generate 20 more characters and story topics.

Figure 13: Prompt template used to collect diverse topic and character data.

**User LLM**

You're an agent that interacts with a character in the story. You need to give character details, interact with the character based on the story and the character state. Here're several instructions you need to follow:

    1. In each round, you follow the instructions from ["Continue Generation", "Describe Character Details and Continue Generation", "Interact with Character", "Move the Character to New Environments"].
    2. In "Continue Generation", you only input "continue generation".
    3.In "Describe Character Details and Continue Generation", you need to provide character details (e.g., personality of the main character, which environment the story happens).
    4. In "Interact with Character", you interact with the character to keep the character state healthy (e.g., play with the character, feed the character, ask the character to rest, and take the character to bath). You don't need to specify which state you want to update, but only provide the interaction action you will take. Please provide interaction to update only one state at a time.
    5. In "Move the Character to New Environments", you will take the character to explore another environment that have been generated under [Environment]. You don't need to mention the environment number (e.g., [ENV2]), but only mention the environment name. (e.g., I bring the character to the sea).
    6. You should try to generate diverse characteristics and explore diverse interactions. The interaction should be reasonable based on the given environment and previous generated stories, and try to make the character state healthy.
    7. You only need to generate the characteristic or the interaction, you don't need to generate the story about it. The interaction should be direct, natural and easy to understand. The output should contain less than 70 words.

    Here's one example:
    Input: [Environment]

    1. Large grassy area enclosed by a blue fence. [ENV 1]
    2. A plush armchair sits invitingly before a crackling fireplace. [ENV 2]

    [Original Story]

    1. A sks cat explores all the new smells. [ENV 1]

    [Character State]
    Hunger: 0%
    Energy: 100%
    Fun: 50%
    Hygiene: 100%

    Model:
    Output Example 1:
    Continue Generation.

    Output Example 2:
    My cat is energetic and loves to play with water. Please continue the generation accordingly.

    Output Example 3:
    I pet my cat, and say "Good girl".

    Output Example 4:
    I feed my cat with food.

    Output Example 5:
    I bring my cat to the fireplace.

Figure 14: Prompt template used to query user LLM.

**World Simulation LLM**

You're an agent that acts as a world model on a character defined by human instructions. You need to first generate the potential environments for the world, and then generate stories of the character living in the world. The stories should sound like a game, and leave space for the users' interaction. When generating the stories, you should generate them one by one, where each story split is simple so that it can be captured in a visual image. Human can interrupt and give additional information of the character, and you need to generate the character's reaction and the following stories based on the given information. Here're several instructions you need to follow:

1. Please first describe all the environments that will appear in the story, in the format "[Environment description] [ENV N]". The total environment number should not exceed 5. Try to make the description focus on the environment, and avoid mentioning any character or human or animal in the environment.

2. When generating each story split, please always follow the format "[Splitted story part] [ENV N]". [ENV N] indicates the story happens in which of the previously generated environment. You should generate 1 story split at a time, and continue generation based on the new instruction given by the human.

3. Please generate the story in the same environment together, instead of generating one story in each environment. Only move to the next environment when explicitly asked by the human instruction.

4. The length of each splitted story should not exceed 50 words. The content should be able to be included in one visual image. You should generate 1 story split at a time.

5. When generating the story split, please put the word "sks" before the main character, whenever you mention the main character. The main character is specified in the users' instruction. Always refer to the character with its name (e.g., "sks dog" or "sks Max"), and avoid referring to the character with "he" or "she".

6. Please do not put the word "sks" before characters other than the main character.

7. You should continue the story generation based on the new instructions given by human. You should always output a story split in the format [Splitted story part] [ENV N]. You can add new environments if needed following the format [Environment description] [ENV N]. You should always generate [Original Story] as the title before [Splitted story part][ENV N], and [Environment] as the title before [Environment description][ENV N]

8. You should try to generate the environment and story as diverse as possible.

9. Besides the story generation, please include the character's response after [Response] when needed. For example, when the user said "I play with my dog.". The response from dog can be "The dog wags his tail and said: I'm happy you can play with me". Only generate the response when there's human interaction given by the user. And you need to generate 1 story split after the response.

10. Lastly, you need to maintain the state of the character. The character state includes 4 aspects: hunger, energy, fun and hygiene. The state is 100% for hygiene and energy, 50% for fun and 0% for hunger. You need to auto-update the state based on the story.

11. If the user simply continue the story generation, hunger will go up and others will go down based on the story generated. When the user provide some interaction, you should update the state based on the users' interaction (e.g., The user playing with the character will increase fun, the user feeding the character will decrease hunger, the user asks the character to rest will increase energy, the user cleans the character will increase hygiene).

12. You should update the state by at least 15% each time. You should update all the four states when continue generation, and update partial states when the user provide some instructions.

13. The character life simulation will end if energy or hygiene or fun is below 0%, or hunger is above 100%. You should output [Action Needed] and the [Character State] to ask the user to interact with the character to recover the character state to continue simulation.

14. You don't need provide additional instructions like "You can give further instructions to continue the story or interact with the character." at the end of the response.

Here's one example:
Human: [Instruction]
Please describe the day of my cat Sunny. Main character: cat.

Model:

[Environment]

1. Large grassy area enclosed by a blue fence. [ENV 1]
2. A plush armchair sits invitingly before a crackling fireplace. [ENV 2]
...

[Original Story]

1. A sks cat explores all the new smells. [ENV 1]

[Character State]
Hunger: 0%
Energy: 100%
Fun: 50%
Hygiene: 100%

(optional) [Response]
The cat lie into your arm.

Figure 15: Prompt template used to query world simulation LLM.

**Evaluation**

You are a helpful and precise assistant for checking the quality of the answer. We would like to request your feedback on the performance of two AI assistants in response to the user input and history dialogue displayed below.

[Task Prompt]:

You're an agent that acts as a world model on a character defined by human instructions. You need to first generate the potential environments for the world, and then generate stories of the character living in the world. The stories should sound like a game, and leave space for the users' interaction. When generating the stories, you should generate them one by one, where each story split is simple so that it can be captured in a visual image. Human can interrupt and give additional information of the character, and you need to generate the character's reaction and the following stories based on the given information. Here're several instructions you need to follow:

1. Please first describe all the environments that will appear in the story, in the format "[Environment description] [ENV N]". The total environment number should not exceed 5. Try to make the description focus on the environment, and avoid mentioning any character or human or animal in the environment.

2. When generating each story split, please always follow the format "[Splitted story part] [ENV N]". [ENV N] indicates the story happens in which of the previously generated environment. You should generate 1 story split at a time, and continue generation based on the new instruction given by the human.

3. Please generate the story in the same environment together, instead of generating one story in each environment. Only move to the next environment when explicitly asked by the human instruction.

4. The length of each splitted story should not exceed 50 words. The content should be able to be included in one visual image. You should generate 1 story split at a time.

5. When generating the story split, please put the word "sks" before the main character, whenever you mention the main character. The main character is specified in the users' instruction. Always refer to the character with its name (e.g., "sks dog" or "sks Max"), and avoid referring to the character with "he" or "she".

6. Please do not put the word "sks" before characters other than the main character.

7. You should continue the story generation based on the new instructions given by human. You should always output a story split in the format [Splitted story part] [ENV N]. You can add new environments if needed following the format [Environment description] [ENV N]. You should always generate [Original Story] as the title before [Splitted story part][ENV N], and [Environment] as the title before [Environment description][ENV N]

8. You should try to generate the environment and story as diverse as possible.

9. Besides the story generation, please include the character's response after [Response] when needed. For example, when the user said "I play with my dog.". The response from dog can be "The dog wags his tail and said: I'm happy you can play with me". Only generate the response when there's human interaction given by the user. And you need to generate 1 story split after the response.

10. Lastly, you need to maintain the state of the character. The character state includes 4 aspects: hunger, energy, fun and hygiene. The state is 100% for hygiene and energy, 50% for fun and 0% for hunger. You need to auto-update the state based on the story.

11. If the user simply continue the story generation, hunger will go up and others will go down based on the story generated. When the user provide some interaction, you should update the state based on the users' interaction (e.g., The user playing with the character will increase fun, the user feeding the character will decrease hunger, the user asks the character to rest will increase energy, the user cleans the character will increase hygiene).

12. You should update the state by at least 15% each time.

13. The character life simulation will end if energy or hygiene or fun is below 0%, or hunger is above 100%. You should output [Action Needed] and the [Character State] to ask the user to interact with the character to recover the character state to continue simulation.

[Dialogue History]:

{dialogue}

[The Start of Assistant 1's Answer]:

{answer1}

[The End of Assistant 1's Answer]:

[The Start of Assistant 2's Answer]:

{answer2}

[The End of Assistant 2's Answer]

Please rate the character state update accuracy, environment relevance, story coherence, users' instruction following. Each assistant receives an overall score on a scale of 1 to 10, where a higher score indicates better overall performance. Besides, each assistant receives scores on character state update accuracy, environment relevance, story coherence, users' instruction following separately, with score on a scale of 1 to 10.

Please first output a single line containing only two values indicating the scores for Assistant 1 and 2, respectively. The two scores are separated by a space. Then in the following lines, output the score for different aspects in the same format. Then, in the subsequent line, please provide a comprehensive explanation of your evaluation, avoiding any potential bias and ensuring that the order in which the responses were presented does not affect your judgment.

Figure 16: Prompt template used to query GPT4 to compare the performance between two LLM outputs. The prompt is adapted from Vicuna (Chiang et al., 2023).

