# OpenReview forum: "Unbounded: A Generative Infinite Game of Character Life Simulation"
_ICLR.cc/2025/Conference — ICLR 2025 Poster_

### Official Review · Reviewer_Nnkj · 2024-11-03

**Soundness:** 3
**Presentation:** 3
**Contribution:** 3
**Rating:** 6
**Confidence:** 4

**Summary:**

This work combines various aspects of Generative AI to develop a LLM-based game engine. The design principles refer to open-world, continuous interaction games with limited simultaneous characters but significant opportunities for situation generation and interaction. Capitalizing upon recent research in the field, the system introduces a number of technical innovations both in terms of architecture (in the form of a dynamic regional prompt adapter) and in term of implementation (distillation of Small LLM). With an existing implementation, the paper includes elements of comparative evaluation,  both quantitative and qualitative.

**Strengths:**

This is an ambitious piece of work, tackling difficult technical issues in balancing generation and interaction, as well as various training and fine-tuning protocols. It appears to have innovated on a number of aspects, for instance the dynamic mask to balance character and environment generation. It gives some interesting, detailed, insights such as those of Figure 4. The improvement over IP-Adapter thus appears more than incremental. Another strong points is the Small LLM distillation framework.
The comparative LLM evaluation has staged an appropriate number of LLM, capturing size and diversity.
Finally, the paper demonstrates a good command of recent relevant work and compares its approach to really recent alternative methods (2023, 2024), which is welcome.
Having considered the authors' rebuttal and additional data, I have raised my score for contribution to 3 ('good').

**Weaknesses:**

From a fundamental perspective, there is a lack of awareness of some game design issues. The introduction is relatively naïve considering how these issues are addressed in game/play theory (Caillois, Huizinga). With the fine line between simulation and entertainment, and notwithstanding the reference to The Sims and Tamagotchi, contemporary audiences tend to have gameplay expectations that can only be met with sustained and meaningful (in the narrative sense) interactions. This feeds into the overall perspective of the evaluation methods which in the paper are primarily focused individual ‘event’ generation. While sustained UX evaluation may be at odds with the system’s current level of maturity, some of this could have been addressed through even slightly longer exchanges or chains of events construed as minimal narrative experience (which is claimed in the abstract but does not really materialize in the examples). The small examples given, including on Figure 1, do not convey a very strong sense of gameplay.

There is no evidence that the number of user-simulator examples is at all sufficient for the intended purpose of multi-topic data collection. Nor is there any discussion of some hierarchical organization of topics that would come close to scenarios, game genres or elementary forms of plot backbones.

Despite the statement on replicability, the level of implementation details is unclear. Implementation details are rather cursory and uneven with some in-depth details but no systematic discussion, and minor duplications from other sections.

Evaluation has both positive (see above) and negative aspects.
Unless I am mistaken, Table I shows a couple of inconsistencies for results.
On Environment Consistency, how can 0.322 be the best score when it is in-between 0.381 and 0.257? How can 0.675 be the best score in-between 0.595 and 0.832?
It is difficult to find the Qualitative component of evaluation very convincing in the absence of a more systematic approach for generating test situations. From a similar perspective, the example Prompts do not contain advanced mechanisms for controlling response, or specific examples. One would thus wonder how LLM response can be kept consistent across runs, and what type of statistical sampling has been associated to LLM use?

**Questions:**

To which extent are results from Figure 6 consistent with results from Table I?
How have you ensured the quality and fairness (for evaluation) of the text prompts sample?
How can the current approach maintain narrative consistency over a number of interactions?

---

> ### Author Response · Authors · 2024-11-22
> **Rebuttal**
>
> >**Discussion of game design related work**
>
> Thank you for pointing this out. We have added discussions on how our work aligns with previous game design literature in both the introduction and related work sections (highlighted in red text in the updated PDF).
>
> >**Long gameplay example**
>
> We include one gameplay example (w/ 15 rounds of user interactions) in Appendix A. This example demonstrates that our game engine can produce a game featuring coherent narratives, consistent mechanics, consistent characters and background, and expansive user interactions.
>
> >**Evaluation of LLM response**
>
> We did not enforce consistency in the LLM's responses across runs, as allowing the LLM to generate varied responses enhances the potential for unbounded interaction experiences during gameplay. To address your concern, we conducted further evaluations using greedy decoding, which constrains the output to remain the same for identical inputs. The results are presented in the table below. We observed that greedy decoding outperforms GPT-4o when evaluated using GPT-4 as the judge. Additionally, we conducted three more runs using sampling (samples 2, 3, and 4) and included the results here. Despite the variations in responses across different samples, all outputs maintain high quality under the evaluation criteria.
>
> | Model | Overall |  | State Update |  | Environment Relevance |  | Story Coherence |  | Instruction Following |  |
> | :---- | :---- | :---- | :---- | :---- | :---- | :---- | :---- | :---- | :---- | :---- |
> |  | Ours | GPT4o | Ours | GPT4o | Ours | GPT4o | Ours | GPT4o | Ours | GPT4o |
> | Greedy | **7.93** | 7.44 | **7.83** | 7.20 | **8.27** | 7.62 | **8.04** | 7.62 | **8.05** | 7.60 |
> | Sample1 | 7.68 | **7.76** | 7.66 | **7.69** | 8.10 | **8.20** | 7.82 | **7.95** | 7.82 | **7.85** |
> | Sample2 | **7.50** | 6.94 | **7.40** | 6.69 | **7.76** | 7.08 | **7.60** | 7.08 | **7.64** | 7.04 |
> | Sample3 | **7.87** | 7.38 | **7.89** | 7.12 | **8.24** | 7.68 | **7.92** | 7.52 | **8.01** | 7.51 |
> | Sample4 | **7.61** | 7.16 | **7.56** | 6.90 | **7.90** | 7.47 | **7.69** | 7.28 | **7.72** | 7.30 |
>
>
> >**Evaluation of image generation quality**
>
> We apologize for incorrectly bolding the best scores in the previous version, and we have corrected this in the updated paper. First, we want to highlight that while IP-Adapter achieves a higher environment consistency score, it often directly reconstructs the original environment image, neglecting the inclusion of the character. This is reflected in its high environment consistency score and extremely low character consistency score, as shown in Table 1 in the main paper.
>
> Second, regarding the evaluation of prompt quality, we conducted our evaluation on a dataset of 5k prompts spanning 100 diverse environments, offering a comprehensive test of the models’ generation capabilities across varied scenarios. In contrast, prior work, such as StoryDiffusion, evaluates on only 2k collected examples.
>
> Lastly, as suggested, we are actively collecting human annotations to compare our approach with baseline methods in maintaining environment consistency and character consistency. We will share the results of these human evaluations as soon as they are available.
>
> >**Narrative consistency**
>
> We would like to clarify that we evaluated story coherence based on the interactions between the user and the game engine (as shown in Table 3 of the main paper), which serves as an assessment of narrative consistency. Specifically, when using GPT-4 as a judge to compare responses from two models, we explicitly instruct GPT-4 to provide scores for story coherence (i.e., narrative consistency) for each response.

---

> > ### Author Response · Authors · 2024-11-26
> > **Human Evaluation on Image Generation**
> >
> > >**Human Evaluation on Image Generation**
> >
> > We present a human evaluation to assess the performance of our dynamic regional IP-Adapter in maintaining environment consistency, character consistency, and semantic alignment with the text prompt. Specifically, we sample 50 character-environment-text prompt triplets from our image evaluation benchmark and generate corresponding images using our approach and StoryDiffusion. Four human annotators were tasked with evaluating the images across three dimensions: character consistency, environment consistency, and semantic alignment. Given that Unbounded functions as a character life simulation game, ensuring the presence of the character in the image is critical. To emphasize this, annotators were instructed to disfavor images where the character is absent. In such cases, the image is also rated lower in terms of environment consistency and semantic alignment. The user interface used for collecting human annotations is shown in Appendix in the updated PDF. To minimize bias, the order of images generated by the two models was randomized, ensuring that annotators were unaware of the model responsible for generating each image. As shown in the Table, our evaluation demonstrates that the dynamic regional IP-Adapter significantly outperforms StoryDiffusion in maintaining character consistency, environment consistency, and semantic alignment with the text prompt.
> >
> > | Criteria                | Win  | Lose | Tie  |
> > |-------------------------|------|------|------|
> > | Character Consistency   | 120  | 38   | 42   |
> > | Environment Consistency | 108  | 42   | 50   |
> > | Semantic Alignment      | 115  | 46   | 39   |

---

> > ### Comment · Reviewer_Nnkj · 2024-11-27
> > **Rebuttal**
> >
> > Thank you for having clarified several aspects of the paper. I have raised my contribution score to '3'.
> > However, I am not inclined to raise my total score to 8 in view of the non-standard approach of narrative which has been adopted. A '7' score would have been perfect but does not seem to be an option this year.

---

> > > ### Author Response · Authors · 2024-11-27
> > >
> > > Thank you for your thoughtful feedback and for raising the contribution score to 3. We sincerely appreciate the time you've taken to review our work carefully and provide detailed comments throughout this process. We understand your perspective that a 7 would be the ideal score to reflect your assessment of our work, and we appreciate you sharing this with us.
> > >
> > > We understand that the scoring system this year presents some new constraints. We wanted to consider the possibility of whether you feel the 7 score you had in mind aligns more closely with the 'accept' description of the “8” score than with the ‘marginally above the acceptance threshold’ of the “6” score. We believe your detailed review and writing is the most important element in evaluating the work, but we know from past experience that final scores can greatly affect the fate of a paper and are important for the Area Chair's decision process, and we deeply appreciate your thoughtful consideration of this matter. Thank you again for engaging with our work in such a detailed manner.

---

### Official Review · Reviewer_4EBp · 2024-11-04

**Soundness:** 2
**Presentation:** 3
**Contribution:** 2
**Rating:** 6
**Confidence:** 4

**Summary:**

The authors present "UNBOUNDED" a virtual pet game based on LLMs. The authors present two novel technical elements to support this game: a regional IP-Adapter for character-environment consistency and a domain-specific distillation approach for finetuning a smaller LLM to allow for real time gameplay. The authors then present experiments comparing their approach to baselines in terms of visual quality and consistency, and overall game quality as scored by GPT-4o.

**Strengths:**

The main strengths of the paper are (1) the originality of entirely basing a virtual pet game on LLMs, (2) the significance of the two primary technical contributions to figure coherent image generation and real time LLM application uses, and (3) the quality of the image portion of the evaluation/experiments. The paper is also overall well-written, especially in terms of the technical aspects. These primarily justify the positive aspects of my above scores.

**Weaknesses:**

The paper has two major weaknesses in its present draft, the way it positions itself in terms of prior work and the evaluation.

### Prior Work

The authors do not appear to have engaged with the field of technical games research at all. This is a shame, as their game can be understood as an AI-based game [1] or specifically a NN game [2]. More broadly, the authors' work in terms of generation would fit into the Procedural Content Generation paradigm [3]. There is a 3+ decades long history of work in this space that is relevant to the authors' work [4], with significant recent work at the intersection of LLMs and game generation [5], and it is crucial that the authors update the introduction and related work to situate their work within this field.

### Evaluation

The authors primarily evaluate their work in terms of image quality and I have no major concern with these results (though the DreamSim results seem to incorrectly attribute the best score to their system). However, I have major concerns with the "Evaluation of LLM Generations", which is functionally the closest thing presented to a holistic evaluation of UNBOUNDED. Firstly, the norm within the area would be to make use of a user study for evaluation purposes [6,7]. Particularly since the authors make such an effort to allow for real time control, this would seem to follow naturally. Secondly, the evaluation is somewhat concerning due to the training process. Specifically, since the distillation approach makes use of GPT-4 to simulate user interaction data and the authors sample for uniqueness/variety, its fairly likely that the newly sampled user interaction data approximations will be very similar to these. Further, the largest model the authors compare to is again a GPT-4 variant. Given that GPT-4 is being used for both generation of data, as the system, and as the evaluation, there's a clear risk for bias. This could be addressed by any number of ways, including automated metrics for measuring game quality developed in prior game generation-like work [8,9]. But at present, it is difficult as an outside reader to get a sense of the actual quality of UNBOUNDED as a game. Failing this, it might be beneficial to include longer examples of gameplay in the supplementary materials at least.

### Figures

While these are not of major concern, I wanted to note two aspects of figures that I felt could be improved. First, the authors make use of a large number of fonts in their figures, and their primary font is a bit difficult to read. I'd suggest updating the figures to use a single, more legible font. Second, Figure 4 is largely superfluous. If the authors wish to retain it, it might be a better fit for an appendix.

1. Treanor, M., Zook, A., Eladhari, M. P., Togelius, J., Smith, G., Cook, M., ... & Smith, A. (2015). AI-based game design patterns.
2. Zhu, J., Villareale, J., Javvaji, N., Risi, S., Löwe, M., Weigelt, R., & Harteveld, C. (2021, May). Player-AI interaction: What neural network games reveal about AI as play. In Proceedings of the 2021 CHI Conference on Human Factors in Computing Systems (pp. 1-17).
3. Shaker, N., Togelius, J., & Nelson, M. J. (2016). Procedural content generation in games.
4. Pell, B. (1992). METAGAME: A new challenge for games and learning.
5. Gallotta, R., Todd, G., Zammit, M., Earle, S., Liapis, A., Togelius, J., & Yannakakis, G. N. (2024). Large language models and games: A survey and roadmap. arXiv preprint arXiv:2402.18659.
6. Anjum, A., Li, Y., Law, N., Charity, M., & Togelius, J. (2024, May). The Ink Splotch Effect: A case study on ChatGPT as a co-creative game designer. In Proceedings of the 19th International Conference on the Foundations of Digital Games (pp. 1-15).
7. Guzdial, M., & Riedl, M. O. (2021). Conceptual game expansion. IEEE Transactions on Games, 14(1), 93-106.
8. Khalifa, A., Green, M. C., Perez-Liebana, D., & Togelius, J. (2017, August). General video game rule generation. In 2017 IEEE Conference on Computational Intelligence and Games (CIG) (pp. 170-177). IEEE.
9. Guzdial, M., & Riedl, M. (2018, September). Automated game design via conceptual expansion. In Proceedings of the AAAI Conference on Artificial Intelligence and Interactive Digital Entertainment (Vol. 14, No. 1, pp. 31-37).

**Questions:**

1. How do the authors position their work in the field of technical game design?
2. What do longer gameplay examples look like?
3. How do the authors respond to my concerns around their "Evaluation of LLM Generations"?

---

> ### Author Response · Authors · 2024-11-22
> **Rebuttal**
>
> >**Prior work discussion**
>
> Thank you for your suggestion regarding prior works in game design research. We have updated our submission to include discussions of game design prior work in both the introduction and related work sections (highlighted in red in the updated PDF). This is our attempt to position our interdisciplinary work that has innovations in generative vision and language domains within the field of AI-based game design.
> Compared to previous work, our primary differentiating contributions are as follows:
>
> (1) Generating all game mechanics, graphics, characters, environments, and narrative using generative models. No logic or assets are left to traditional game engines.
>
> (2) Adding personalization with custom characters and story arcs. Specifically, we have state-of-the-art innovations in character customization and character/background preservation in the context of visual story generation.
>
> (3) Achieving real-time interactivity through innovations in vision and language.
>
> We hope for a holistic evaluation of these contributions and thank the reviewer for the insightful comments and suggestions which we made our utmost attempt to integrate in the current version.
>
>
> >**Evaluation of LLM as game engine with GPT-4 as a judge**
>
> First, we would like to clarify that utilizing GPT-4 as a robust evaluator for comparing the outputs of two models is a widely recognized evaluation paradigm in the field of LLMs and in very prominent recent LLM work. For instance, Vicuna [1] fine-tunes Llama on ShareGPT data (human conversation data with ChatGPT, a GPT-4 series model) and evaluates model performance using GPT-4. Similarly, WizardLM [2] collects complex instruction-tuning data via the ChatGPT API, fine-tunes Llama to match GPT-4 performance, and evaluates using GPT-4. This approach is broadly adopted in other well-known works (e.g., Camels [3], Parrot [4]), and some benchmarks (e.g., MT-Bench [5]) also leverage GPT-4 as the evaluator to assess both GPT-4 series models and other LLMs.
>
> Additionally, we want to emphasize that we collect data using GPT-4o and evaluate it with GPT-4, with GPT-4 being a stronger model than GPT-4o. This ensures that the evaluation is conducted by a more capable model, rather than using the same model for both data generation and evaluation.
>
> Lastly, as per your suggestion, we are currently conducting human evaluations comparing GPT-4o with our distilled LLM. We will update the results here once the human evaluations are completed.
>
> >**Evaluation with automatic metrics**
>
> Our goal is to create a generative infinite game where all rules and character behaviors evolve through interactions between the user and the LLM-based game engine. Due to the unbounded possibilities for interactions and open-ended responses, constructing an evaluation dataset to define a "correct" response is challenging, as there is no single correct answer. For these open-ended responses, consistent with practices in LLM research[1,2,3,4,5], we use a stronger LLM (GPT-4) as a judge to evaluate which LLM engine performs better.
> In this automated evaluation process using GPT-4, we assess performance across multiple dimensions:
>
> (1) Accuracy in maintaining character state based on user inputs.
>
> (2) Relevance of the auto-generated environment to the game topic provided by the user.
>
> (3) Story coherence as the game unfolds through user-character interactions.
>
> (4) Adherence of the character to user instructions.
>
> Lastly, as suggested, we have included a qualitative example of long-play from our model's in Appendix Section A (please refer to the updated PDF). This example demonstrates that our game engine can produce a game featuring coherent narratives, consistent mechanics, consistent characters and background, and expansive user interactions.
>
> >**Figures**
>
> Thanks for your suggestion on Figures. We’ve moved Figure 4 to Appendix.
>
> [1] Vicuna: An Open-Source Chatbot Impressing GPT-4 with 90%* ChatGPT Quality
>
> [2] WizardLM: Empowering Large Language Models to Follow Complex Instructions
>
> [3] How Far Can Camels Go? Exploring the State of Instruction Tuning on Open Resources
>
> [4] Parrot: Enhancing Multi-Turn Instruction Following for Large Language Models
>
> [5] Judging LLM-as-a-Judge with MT-Bench and Chatbot Arena

---

> > ### Comment · Reviewer_4EBp · 2024-11-23
> > **RE: Rebuttal**
> >
> > Thanks to the authors for their effort in updating the paper and their responses to the issues I identified.
> >
> > The paper is definitely improved in terms of its coverage of prior technical games work. I would still quibble with the claim that the authors "introduce the concept of a generative infinite game" as I think that's an overly broad term for what the authors have done. "Generative infinite game" could just mean "roguelike". I would suggest tempering this claim or acknowledging this criticism in the paper.
> >
> > I acknowledge that there is a large amount of work using LLMs as evaluators. However, there has been no work demonstrating that an LLM evaluator will align with human judgements in this domain. As such, I still would argue that this is not ideal for the purposes of this work. I am pleased to hear that the authors plan to run/are running a human subject study and I look forward to those results!
> >
> > For now, weighing the above improvements and my remaining concerns, I have increased my Presentation rating from fair to good, and increased my overall score from 5 to 6.

---

> ### Author Response · Authors · 2024-11-26
> **Human Evaluation on LLM as Game Engine**
>
> Thank you for your thoughtful feedback and for taking the time to review our rebuttal. We are pleased to hear that the revisions addressed some of your concerns. Additionally, we have conducted a human evaluation of the LLM as a game engine and included the results below:
>
> >**Human Evaluation on LLM as Game Engine**
>
> We conduct a human evaluation to assess the performance of our distilled LLM as a game engine. Specifically, we collect human annotations on 10 game topics, with each topic comprising five rounds of user-model interactions, resulting in a total of 50 annotated examples. Model responses are generated using either our distilled small LLM or GPT-4o. Four human annotators evaluate across five categories, which are the same as those used in GPT-4's automatic evaluation: overall score, state update accuracy, environment relevance, story coherence, and adherence to user input instructions. The user interface employed for collecting human annotations is detailed in the Appendix in the updated PDF. As shown in the table below, the human evaluation demonstrates that our distilled small LLM achieves performance comparable to GPT-4o across all evaluated categories.
>
> | Criteria              | Win | Lose | Tie  |
> |-----------------------|-----|------|------|
> | Overall Score         | 53  | 54   | 93   |
> | State Update          | 41  | 48   | 111  |
> | Environment Relevance | 32  | 31   | 137  |
> | Story Coherence       | 30  | 36   | 134  |
> | Instruction Following | 29  | 29   | 142  |

---

> > ### Comment · Reviewer_4EBp · 2024-11-27
> > **Re: Human Evaluation on LLM as Game Engine**
> >
> > Thanks to the authors for the quick response and additional results! It's definitely nice to see that the results appear to be comparable according to human judgements. I also appreciate the inclusion of the interface in the appendices and the details around the methodology. However, I have additional questions related to the methodology for this experiment.
> >
> > 1. Did the authors seek ethics approval or review for this experiment?
> > 2. How were the annotators recruited?
> > 3. How were the annotators compensated, if at all?
> > 4. What domain knowledge of AI and games did the annotators have?
> >
> > I ask these questions as while the choice to randomly present examples from the different approaches (which I assume was done here given the text of the appendix) would help reduce the risk of bias, these all could lead to potential biases in the results. As such, it's important to clarify these details.

---

> > > ### Author Response · Authors · 2024-11-27
> > >
> > > Thanks for the quick response! Besides the efforts in randomizing the response order to mitigate bias in annotation, here are the additional details requested:
> > >
> > > 1. The experiment went through our institution’s review and approval process. Furthermore, the study has been assessed to have minimal risk for participants and participant response data is anonymized. We have obtained full informed consent from all participants in the study. Participants were unpaid volunteers.
> > >
> > > 2. We recruit four annotators with a randomized outreach approach formulated with neutral and inclusive messaging in order to reduce selection bias. The surveyed population is a student population with graduate or post-graduate education level. The final group includes two students in Computer Science, one student in Economics, and one student in Finance. We did not collect specific responses on familiarity of games and AI, though, given the participant backgrounds these would tend to include some variety.

---

> > > > ### Comment · Reviewer_4EBp · 2024-11-28
> > > > **Re: Official Comment by Authors (Human evaluators)**
> > > >
> > > > Thanks to the authors for the additional information! This does help remove any ethical concerns I had. However, given that the raters are non-experts, it would be more typical to have at least >10 responses to ensure that the participants ratings were representative of the general population. As such, I will retain my first score improvement.

---

### Author Response · Authors · 2024-11-22
**General Response**

We thank all the reviewers for their thoughtful feedback. We’re glad that they acknowledge our originality of entirely basing a virtual pet game on LLMs (reviewer 4EBp), and acknowledge our two main technical contributions in coherent image generation and real-time LLM application uses (reviewer 4EBp, Nnkj), and recognize that our paper has good image generation evaluations (reviewer 4EBp) and compares to really recent alternative methods (reviewer Nnkj). We address their questions below and have updated the submission paper accordingly, and hope for a re-evaluation of the paper score if their concerns have been addressed.

---

### Meta-Review · Area_Chair_pJNT · 2024-12-23

**Metareview:**

The paper presents an interesting experiment in creating a generative game where the core "game engine" is based on LLMs and diffusion models. While obviously there is a long way to go, there are interesting things to learn from the paper. The "core technical" contribution of the paper is perhaps the regional IP-adapter used to ensure that placement of characters in images work, but the bigger contribution is the system as a whole. It would have been nice with a proper user study to understand how people play with the system, but... next paper?
The first version of this paper was seriously lacking in its related work section, essentially ignoring the most relevant research community, but this has been much improved in the updated version.

**Additional Comments On Reviewer Discussion:**

The authors engaged well with the reviewers and updated their paper appropriately.

---

### Decision · Program_Chairs · 2025-01-22

Accept (Poster)